# Are They Reporting the Right Thing and Are They Doing It Right?—A Measurement Maturity Grid for Evaluation of Sustainability Reports

**Mathias Cöster, Gunnar Dahlin * and Raine Isaksson** 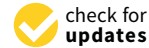

Department of Civil and Industrial Engineering, Uppsala University, 752 36 Uppsala, Sweden;
mathias.coster@angstrom.uu.se (M.C.); raine.isaksson@angstrom.uu.se (R.I.)

**\*** Correspondence: gunnar.dahlin@angstrom.uu.se

**Abstract:** An accessible way to monitor company sustainability, is to study sustainability reports. In spite of wide adherence to the extensive Global Reporting Initiative standards, sustainability reports still vary considerably regarding how well these are integrated and used. The purpose of this paper is to present and test a maturity grid for sustainability reports assessment that enables critical stakeholder needs analysis of sustainability reports. Based on a stakeholder needs perspective we argue that the right thing in a sustainability report means reporting in the entire value chain for main sustainability impacts. Doing this right means having externally set targets for main sustainability impacts, using relevant absolute and relative indicators, and having an easy to read report presenting main performance compared to targets for a period of at least seven years. Some 50 sustainability reports from Swedish companies in various industries were collected and assessed using the maturity grid. Results indicate that the maturity grid is usable, but that the sustainability report assessment still is difficult, and that variability of the assessments are high. Furthermore, the observed indicative levels of measurement maturity in organizations are low suggesting that most companies still are struggling with understanding what sustainability means to them.

**Keywords:** maturity model; maturity grid; measurement maturity; stakeholder focus; sustainable development; sustainability reports

## 1. Introduction

Sustainability reporting is an important part of sustainability performance management [1–3]. According to e.g., Clément Roca and Searcy [4] and Lozano et al. [2], the Global Reporting Initiative Guidelines (GRI), which in 2016 have been converted to standards, seem to be the dominating reporting structure. The standards GRI 101, 102, and 103 describe sustainability report content and the GRI 200, 300, and 400 series contain three sets of standards for sustainability indicators. Each standard has in turn several topic specific disclosures, which brings the total amount of indicators to about 100. Altogether, this wide range of standards may be hard to oversee and interpret, which in turn can make it hard for organizations to identify which of them are crucial to address in a sustainability report [5].

Isaksson and Cöster [6] highlight that out of 39 different sustainability reports for companies operating in Sweden, 37 refer or report accordingly to the GRI. Still, only about 20% of these reports includes the entire value chain in which the company is active. This indicates that interpretation of the GRI standards still varies. Additional studies of current sustainability reports support this assumption [5,7,8]. Altogether there still seem to be severe shortages in how sustainability performance is presented. However, there are also studies arguing that sustainability reports appear to actually indicate sustainability performance [9].

The variation and heterogeneity of sustainability reports indicates several problems. It makes it almost impossible for stakeholders to critically analyze if an organization is doing the right things and if it is doing them right. Based on concepts of Quality Management the right thing in a sustainability report means reporting in the entire value chain for main sustainability impacts [10]. Doing things right means having relevant indicators, externally set targets and a report that is easy to read. The purpose of this paper is therefore to present and test a maturity grid for sustainability reports assessment that enables critical stakeholder needs analysis of sustainability reports.

A total of some 50 sustainability reports collected in Sweden have been assessed by students and researchers using the proposed maturity grid. Results indicate that the maturity grid is usable but that the maturity assessment still is difficult, and that the measurement variability is high. The found level of sustainability maturity is low, indicating that most studied companies (mainly Swedish or operating in Sweden) still are struggling with understanding sustainability from a needs based perspective.

The paper is organized as follows. First a section on sustainability reports and the concept of maturity grids. Then a proposed maturity grid for assessment of sustainability reports is introduced followed by a method section describing how the validity of the grid has been established. Then empirical results and analysis of challenges using the grid are presented. Finally, some conclusions on how the maturity grid may help organizations to develop their sustainability reports by enabling them to analyze if the organization is performing the right things and if they are doing them right.

## 2. Sustainability Reports and Maturity Grids

### 2.1. Doing the Right Things and Doing them Right in a Sustainability Report

The system thinker Russel Ackoff defines effectiveness as doing the right thing and efficiency as doing the things right [11]. One challenge using this perspective when performing a sustainability report is to define what the right thing is. Mura et al. [12] carry out an extensive literature study on the evolution of sustainability measurement research, identifying 712 articles. This results in eight different areas of inquiry, none of which provide any clear help to the question of what the right thing to measure is. One passable road though could be to include the entire value chain [13]. GRI states that: "The organization describes how economic, environmental, and/or social topics relate to its long-term strategy, risks, opportunities, and goals, including in its value chain." The GRI glossary in one of the clarifying notes defines the value chain as: "The value chain covers the full range of an organization's upstream and downstream activities, which encompass the full life cycle of a product or service, from its conception to its end use." The scope guides the content and the identification of main stakeholders and their needs. Furthermore, in the GRI 101 there are four parts describing the content (doing the right thing) of a sustainability report: Stakeholder Inclusiveness, Sustainability Context, Materiality, and Completeness [14]. Doing the right thing in the context of measuring sustainability performance could therefore, based on the GRI guidelines, be defined as identifying main sustainability impacts in the entire value chain. Isaksson [8] argues that doing the thing right should also be viewed from the reader perspective, which highlights the necessity of clarity and comparability. This is provided that the report can be trusted for being accurate, balanced, reliable and timely. Isaksson and Steimle [15] write: "The simple question of an interested reader looking at a sustainability report is how sustainable is the company and how are they improving". A good sustainability report should therefore present the position, direction and sustainability goals of a company, similarly to how financial performance is presented [15–17].

The GRI 101 furthermore describes reporting quality with seven components, which can be seen as indicators of whether an organization is doing things right: Accuracy, Balance, Clarity, Comparability, Reliability, and Timeliness. Delai and Takahashi [18] found based on a study of eight sustainability measurement systems that "There is not a single initiative analyzed that tackles all sustainability issues and in fact there is no consensus around what should be measured and how". The declared limitation

in the study is that it does not provide any guidance to select the sustainability key issues for an organization. Based on this it seems that agreed guidance of what to report could be missing.

## 2.2. Stakeholder Influence on Sustainability Report Content

Silva et al. [19] observe in their literature review that "Stakeholder expectations have not been considered in depth, which is a possible explanation for the dissatisfaction of stakeholders with current sustainability performance measurement and assessment approaches". In this context stakeholders are normally seen as different potential readers of the sustainability report. On an aggregated level the question could be asked though, which are the main stakeholders when considering People, Planet, and Profit in the Triple Bottom Line (TBL) of social, environmental, and financial performance dimensions? This is important to consider since it affects what the right thing to report is. Isaksson et al. [20] propose that the primarily stakeholders are humanity (People) and nature (Planet). Based on this the right thing to report becomes what is important for Planet and People.

Isaksson [10] takes this discussion further and proposes that the Planetary Boundaries framework can be used to identify main Planet needs. For People the UN Sustainable Development Goals (SDGs) are proposed as a way to identify both Planet and People needs [21]. Isaksson [10] suggests that the license to operate for companies (Profit) should depend on their ability to produce People value, while minimizing Planet and People harm. This is contrary to the assumption in, e.g., Delai and Takahashi [18] who argue that focus should equally be on the three dimensions of the TBL. The World Business Council for Sustainable Development (WBCSD) launched in 2000 the expression Eco Efficiency indicating sales value created per ecological harm [22]. However, value again is sales value, not the utility for the customer or in a wider sense, People value.

## 2.3. Maturity Grids Design Enabeling Identification of the Right Things Right

One method useful when identifying if organizations are doing the right things and if they are doing them right, is to establish a maturity grid. This is an analytical tool used in various fields such as process management, project management, software development, risk management and supply chain management [23–25]. Historically maturity grids are an approach for improving and assessing maturity [26]. Here the maturity grid of Crosby (1979) is regarded as the starting point of the maturity concept [27–30]. In a maturity grid different characteristic of behaviors are exemplified in order to find out the current position of the operation within an organization. In order to develop the level of maturity, suggestions for improvement are based on the differences of behaviors between the current and next level of maturity [27].

Correia et al. [31] carry out a literature review on a form of maturity grid (i.e., maturity models) in supply chain sustainability. The approaches in the studied papers are varying and there is no obvious help in these papers regarding identifying the right thing. Most of the approaches are instead based on identifying different enablers such as presence of policies, strategies, or standards. In a well cited paper Baumgartner and Ebner [32] discuss sustainability strategies and maturity grids and they observe that "Although many companies investigate sustainability management and publish sustainability reports, their main focus in this endeavor remains unclear". The authors discuss sustainability aspects and sustainability issues as a prerequisite for proposing 21 aspects based on the TBL to form the basis for a chosen sustainability strategy. This approach suffers of the ambiguity of the word aspect. The proposed aspects are a mixture of cause and effect and are such as innovation and technology, collaboration, and biodiversity. Logically identification of key aspects should start with identifying key sustainability impacts. The word aspect used to form a core part of the vocabulary but GRI has changed its approach and now uses sustainability impact in order to be more precise. The word aspect has hereby disappeared from the GRI glossary.

### 3. Proposal of a Sustainability Report Maturity Grid (SRMG) for Analysis of Sustainability Reports

So, what may be the fundamentals of a sustainability report maturity grid (SRMG) for analysis of sustainability reports? The SMRG that we have chosen as starting point is based on the structure of Crosby's quality management maturity grid (QMGG). The QMGG of Crosby, which contains of six different measurement categories divided into five different levels of organizational behavior—maturity levels: shapes a matrix of total 30 boxes containing a description of typical organizational behavior characterizing an organization performing on a maturity level regarding each measurement category. In the SMRG in comparison to Crosby's QMGG, Isaksson [8] identified three different measurement categories being used in order to assess sustainability performance—scope: stakeholder needs focus and performance and targets. We have based this work largely on previous work presented by Isaksson and Cöster [6] and Isaksson [8] who propose a SRMG which is based on the outside in approach. We will here argue for why we believe that this grid and the ideas presented in it can be used and further developed to assess the level of reporting maturity or quality. The grid does not intend to measure the level of company sustainability but only the sustainability reporting quality. That is, the grid does not intend to assess how sustainable or unsustainable the company is, but only how the reporting quality or reporting quality maturity is. The reporting quality is defined based on doing the right thing right [11]. The grid contains stakeholder needs focus, scope, and performance and targets [8]. The first two criteria describe the right thing and the third doing it the right way. The first criterion for the right thing for the company is to report in the whole value chain for the main sustainability impacts on main stakeholders. This logic is supported by how the GRI standards suggest reporting and [8]. The second criterion for doing the right thing is to focus on the main stakeholders in the value chain. The final stakeholders in the value chain have been identified as People and Planet and as the main stakeholders. This is an interpretation of the Brundtland commission definition of Sustainable Development and Isaksson's study [20]. Doing the thing right is defined as having externally set targets for the main stakeholder needs and reporting them clearly.

These three criteria have been used to assess performance on a 0–5 scale [6,8]. For each measurement category descriptions of characteristics regarding sustainability focus were composed in an increasing maturity scale from 0 to 5 (i.e., nothing found to mature). This shaped a matrix of 18 boxes containing descriptions of typical sustainability characteristics of a sustainability report. A typical scale for maturity reports like in the Capability Maturity Modell (CMM) goes from 1 to 5. It is argued that there are companies that have not started with any work and therefore are not immature, which would be one [8]. When no work has been done the scale therefore logically should start from zero. The scale goes from nothing to perfect reporting of the right thing in the right way. The intermediate levels of 1–4 represent stages of maturity relating to the end points. Isaksson and Cöster [6] test the proposed matrix on 39 sustainability reports, letting students and a researcher assesses all reports. Main conclusions from the study are that in spite of the help that the proposed grid is providing, students still struggle in assessing the reporting maturity correctly. They systematically rate reports higher than the researcher who is also the expert teacher and used as reference. Results do not show any clear correlation between student rating and researcher rating. Isaksson and Cöster [6] review the matrix critically and propose a revised version with four criteria, where the criterion of readability is added to clarify doing the thing right. The four criteria defining the measurement maturity are presented in Table 1 in a matrix together with the six levels (0–5).

**Table 1.** A maturity grid for assessment of sustainability reports.

| Level<br>Statement | 0 | 1 | 2 | 3 | 4 | 5 |
|---|---|---|---|---|---|---|
| **Scope—<br>Using the value chain to defining scope of reporting—doing the right thing** | Value chain has not been identified | Value chain is identified or implied but not clearly defined | Value chain is defined clearly and visualized | Value chain covers clearly the entire supply chain and is used as basis for describing some important stakeholder needs including carbon emissions | Value chain clearly presents a basis for assessing all prioritized stakeholder needs | Value chain and its life cycle are clearly used for all prioritized stakeholder needs |
| **Stakeholder needs (People and Planet)—doing the right thing** | No clear focus on main sustainability impacts. No clear identification of relevant stakeholders or stakeholder needs | Focus on sustainability is mentioned; Some stakeholders and stakeholder needs have been identified | Sustainability focus has been clearly indicated identifying some relevant global stakeholders and their needs with an appropriate materiality analysis | Sustainability has been clearly defined for the organization identifying main stakeholder needs based on UN SDGs and global limits or similar | Clear link from chosen SDGs, planetary boundaries and from chosen regional and local goals to relevant sustainability impacts | Clear priorities for the relevant sustainability impacts |
| **Performance and targets—doing the thing the right way** | No clear indicators and targets | Some targets and corresponding indicators that are mostly self-referential | Some targets covering relevant stakeholder needs with corresponding indicators<br>Some self-referential targets; Absolute values for main indicators | Indicators for main relevant stakeholder needs present the position and its development over at least three years. There are externally defined relevant targets; Some relevant relative indicators are presented | Indicators for relevant impacts including carbon emissions for the value chain present the development over at least five years also indicating how a level of sustainability based on externally defined targets will be achieved; Relative indicators for carbon footprint | KPIs that describe current performance and rate of improvement in relation to externally based targets for all relevant People and Planet stakeholder needs both in absolute and relative terms |
| **Readability of results—doing the thing the right way** | Results mainly in text with no focus on layout | Some efforts for layout highlighting main performance. Easy to read content. | Clear separate section for reporting sustainability or separate and complete sustainability report presenting main results graphically | Easy to read layout including graphs, among them carbon emissions, presenting relevant performance. Large text enabling easy scrolling of entire pages in pdf. | Presentation highlighting relevant results in easy to understand graphical form combining trend and target. | Extensive report with executive summary presenting the situation and performance over time for all relevant KPIs in an easy to read manner |

Maturity in the grid (Table 1) is defined as the level of measurement maturity the sustainability report provides. The main stakeholder needs for Planet are based on the Planetary Boundaries framework and on Steffen et al. [33] that identify climate and loss of biodiversity as main global concerns. For main People stakeholder needs the UN Sustainable Development Goals (SDGs) have been used and poverty has been singled out as the main social issue [10]. The second part of the reporting quality consists of doing the thing right, which is defined with two criteria. The first criterion is having KPIs that describe key sustainability impacts for the main stakeholder needs and having externally set targets for these following the logic of Science Based Targets (SBT) and using planetary thresholds [34,35]. That is, targets should be need based and externally set. The second criterion is report readability which is based on the idea of the interested reader as customer for the report and the GRI standard quality requirement of Clarity.

Carbon reporting has been chosen as an indicator variable. Due to the role of climate change all companies need to identify their effects on it, irrespective if it is low or high. In comparison with effects of the value chain on biodiversity and poverty, reporting carbon emissions, or carbon emission equivalents is relatively easy. There are companies like the IKEA group that in spite of a long and complex value chain are able to present the carbon emissions in the entire value chain fully including all carbon reporting Scopes 1–3 [36]. The Scope 3 emissions which represent up- and downstream emissions are not assured but presented with a best estimate. Thus, how carbon reporting has been dealt with is proposed to be an indicator for the company commitment to sustainability. If what is important and relatively simply to measure is not reported correctly, then there is serious concern for the quality of the entire report.

The SRMG grid is furthermore to be considered as descriptive as it holds criteria and assessment methodology for each maturity level. Based on this, the SRMG in Table 1 is to be interpreted as follows:

In the left column, four different statements are highlighted. Scope, Stakeholder needs (People and Planet), Performance and target, and Readability of results. The first two address the issue if an organization is doing the right thing, while the latter address if they are doing it right. Then follows columns numbered 0–5, indicating increasing levels of sustainability activities presented in the reports.

The text in the boxes gives directions to how a reader of a sustainability report should estimate the degree of maturity that the report represents. Here, the scale is increasing and 0 equal's very low level of maturity, while 5 indicates very high levels of maturity. E.g., if sustainability has been clearly defined for the organization identifying main stakeholder needs, the report has a level of maturity equal to 3.

## 4. Materials and Methods

The method in this paper is inspired by Innovation Action Research (IAR), a method that allows the researcher to combine action research with iterative innovation [37]. Particularly the part of repeating and developing the original idea and presenting it repeatedly in several contexts has been applied. The work started in organizational research with companies asking them to identify main sustainability impacts. These ideas were tested with students in several courses. This resulted in a conference paper with a draft maturity model, which was then developed into an article [8]. A model presented by Isaksson [8] was tested with students from Uppsala University in spring 2018 resulting in a proposed model [6]. The proposed SRMG was then subjected to a pilot teste by a group of Master students studying a course called Managing Sustainability at Uppsala University in fall 2018. The 20 students formed five groups, which each analyzed a sustainability report. The purpose of this was to find out if there were any obvious flaws in the maturity grid. In the next stage, third year bachelor quality management students in a course called Sustainable Business Development at Uppsala University, performed as campus and as distance course in early 2019, were used as readers. These third-year students who previously had done a 10 ECTS course in sustainable development during their first bachelor year (corresponding to 1/6th of a year) were considered as advanced readers and stakeholders with good knowledge into the subject of sustainability. It was judged by the researchers that these

students should be able to read and make an assessment using the SRMG. This method is a repetition of work done by Isaksson and Cöster [6] with a previous version of the maturity grid.

The students were free to choose any company that produced a sustainability report. Students on campus were directed to choose different reports whereas those on distance working independently had no restrictions. When all results where pooled together it showed that there were a number of multiple report reviews, i.e., reviews of the same reports. The choice of companies could be considered random, but with a certain bias towards larger well-known Swedish companies. The students were assigned to analyze the chosen sustainability report and as one part of the analysis they had to use the maturity grid in Table 1. All three authors of this paper rated the sustainability reports chosen by the students independently. In order to also calibrate the author rating before the assessment, five sustainability reports from different types of businesses were studied and rated independently by the authors. The results and differences were then discussed with the purpose of reducing rating variation in the test and to obtain input for improving the maturity grid.

In total 64 student reports were included in the study. This included several multiples. This means that the number of unique comparable reports was reduced to 52 reports. For those reports with several student assessments a student average was calculated, and this average was then compared with the researcher average. All researchers studied 52 different reports. Some of the student reports were later found not be complete, which resulted in a data set of 49 different sustainability reports that were examined by all three researchers and at least one student. The student reports originated from both campus and distance courses in spring 2019 but were dealt with as one sample since there is no reason to believe that there is any systematic difference. The researchers have dedicated about 15–20 min for each sustainability report to be able to do the rating based on the proposed SMRG. The student, home-based tasks have been extensive, and their individual reports have been about 10–20 pages, which indicates a significant amount of work conducted by the student in order to analyze and interpret the sustainability report and judge the level of sustainability maturity based on the proposed SMRG The student reports included review and reporting of the theory based on the study [6] as well as more detailed questions related to the sustainability report, which supposedly required the student to read the report in more detail. The student time for reading the report has not been assessed, but it probably was more than the time used by the researchers.

## 5. Results

### 5.1. Companies Studied

The companies listed in Table 2 were the ones chosen by students and have all been reviewed by the three participating researchers. In the first two columns the name of the company and the business it performs is presented. Then follows a column where the format of the reports is coded as 0 which mean free format, 1 a format inspired by GRI, and 2 a format that is according to GRI. Finally, in the right column a 0 means that it is a solitary sustainability report, while 1 is equal to an integrated financial and sustainability report.

Results from Table 2 confirm previous results of the GRI standards being the dominating framework for sustainability reporting. About 50% of the reports ($n = 25$) are integrated report which means that the sustainability report is presented together with the yearly financial report. This corresponds to a recommendation from the GRI guidelines. Out of the 52 reports studied it was found after comparison with student reporting that some of the student reports were incomplete and therefore could not be used for comparison. This reduced the final number to 49 reports.

**Table 2.** List of companies studied by researchers (*n* = 52).

| Company | Type of Company | Report Format | Report Integration |
|---|---|---|---|
| Akademiska Hus 2017 | Building property management | 2 | 1 |
| Arvid Nordqvist 2017 | Coffee production and sales | 1 | 0 |
| Axfood 2017 | Grocery store company | 2 | 1 |
| Billerud Korsnäs 2017 | Pulp and paper manufacturer | 2 | 1 |
| Björn Borg 2017 | Fashion | 1 | 0 |
| CCEP 2017—Coca Cola EP | Soft drinks production and distribution | 2 | 0 |
| Cloetta 2017 | Chocolate production | 2 | 0 |
| Coca Cola Co 2017 | Soft drinks production and distribution | 2 | 0 |
| COOP 2016 | Grocery store company | 1 | 1 |
| Coop 2017 | Grocery store company | 2 | 1 |
| Coor 2017 | Facilities support service company | 2 | 1 |
| Daniel Wellington 2017 | Fashion | 0 | 0 |
| Destination Gotland 2017 | Shipping | 0 | 0 |
| Ellevio 2017 | Electricity distribution company | 2 | 1 |
| Findus 2017 | Frozen foods production | 1 | 0 |
| Folksam 2017 | Insurance company | 1 | 0 |
| Gina Tricot 2017 | Fashion | 2 | 0 |
| Gotlandshem 2017 | Production and renting of real estate | 0 | 1 |
| H&M 2017 | Fashion | 2 | 0 |
| ICA 2015 | Grocery store company | 2 | 1 |
| ICA 2017 | Grocery store company | 2 | 1 |
| ICA 2018 | Grocery store company | 0 | 1 |
| ICA2018—October–December 2018 | Grocery store company | 0 | 0 |
| KappAhl 2018 | Fashion | 2 | 1 |
| Kicks 2018 | Cosmetics | 1 | 0 |
| Kinnarp 2018 | Furniture | 0 | 0 |
| Kopparbergs Bryggeri 2017 | Brewery | 0 | 0 |
| Lindex 2017 | Fashion | 2 | 0 |
| Länsförsäkringar 2017 | Insurance | 0 | 0 |
| Mimer 2017 | Production and renting of real estate | 0 | 1 |
| Mini Rodini 2017 | Fashion | 1 | 0 |
| Nudie Jeans 2017 | Fashion | 1 | 0 |

**Table 2.** *Cont.*

| Company | Type of Company | Report Format | Report Integration |
|---|---|---|---|
| Oatly 2017 | Food production | 0 | 0 |
| Peab 2017 | Building | 2 | 1 |
| Postnord 2017 | Postal services | 2 | 1 |
| SAS 2017 | Airline company | 2 | 0 |
| Scandic 2018 | Hotel | 2 | 1 |
| Securitas 2017 | Security company | 2 | 0 |
| Skistar 2017/2018 | Ski resort | 2 | 0 |
| SSAB 2015 | Steel producer | 2 | 0 |
| SSAB 2017 | Steel producer | 2 | 1 |
| Stadium 2016–17 | Sports and sports fashion | 2 | 0 |
| Stena Metall 2017/2018 | Metal recycling and production | 2 | 1 |
| Sveaskog 2017 | Forest management and timber production | 2 | 1 |
| Swedavia 2017 | Airport management | 2 | 1 |
| Swedbank 2017 | Bankt | 2 | 1 |
| Södra Cell Mörrum | Pulp production | 2 | 1 |
| Uppsalahem 2017 | Production and renting of real estate | 2 | 1 |
| Vattenfall 2017 | Power company | 2 | 1 |
| Volvo Group 2017 | Manufacturing and sales of busses, trucks and construction equipment | 2 | 1 |
| VPK Packaging 2018 | Packaging and containers | 2 | 0 |
| Västervik Energi och miljö | Energy distribution, waste management, water management, distance heating | 0 | 0 |
|  | According to GRI | 33 | Integrated report—25 |
|  | Based on GRI | 8 | Sustainability report 27 |
|  | Other format | 11 |  |
|  | GRI influence | 79% |  |

## 5.2. Results from Application of the Proposed SRMG

The overall results for the level of measurement maturity of the 49 studied Swedish sustainability reports presented in Table 3 is low. The average for the four criteria has been calculated. The student average is 2.5 and the researcher average is 1.4 on a scale from 0 to 5, see Table 2. Students rate reports systematically higher than researchers. This result is similar to results from [6] where the averages were 2.8 and 1.7, respectively, and the difference comes out as the same 1.1 higher for students as in the current study. The researcher rating shows some variation, but all researchers rate the reports significantly lower than the students.

**Table 3.** Average results from assessing 49 sustainability reports using grid in Table 1.

|  | Scope | Stakeholders | Performance | Readability | Average |
|---|---|---|---|---|---|
| **Researcher 1** | 1.3 | 1.4 | 1.0 | 1.2 | 1.2 |
| **Researcher 2** | 1.2 | 1.6 | 1.3 | 1.5 | 1.4 |
| **Researcher 3** | 1.4 | 1.7 | 1.8 | 1.7 | 1.7 |
| **Researcher average** | 1.3 | 1.6 | 1.4 | 1.5 | 1.4 |
| **Diff Max–Min researcher** | 0.2 | 0.2 | 0.8 | 0.5 | 0.4 |
| **Students average** | 2.1 | 2.4 | 2.6 | 3.1 | 2.5 |
| **Diff Student-Res** | 0.8 | 0.8 | 1.3 | 1.6 | 1.1 |

The difference is lower for scope and stakeholders. The criteria scope, stakeholders and performance were also studied by Isaksson and Cöster [6] with the recorded differences being Scope 1.4, Stakeholders 0.9 and performance 0.9 higher for student compared to researcher. The reduction in the difference for scope could have been affected by an increased focus on the value chain in the student education. The new criterion of readability is the one with the highest difference, which could indicate that the descriptions in the grid are not clear enough.

*5.3. Comparing the Student and Researcher Ratings*

Average ratings from researchers and students were compared to find out the correlation between researcher and student ratings. These results are presented in Figure 1.

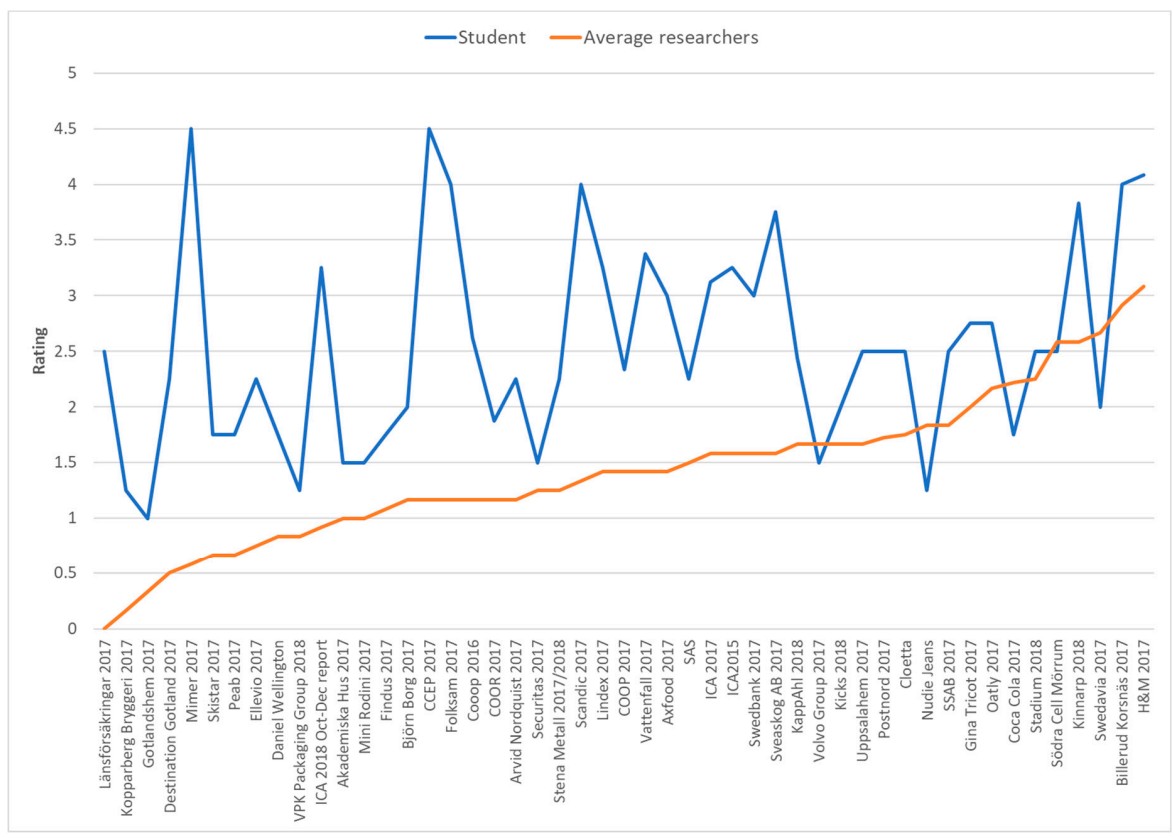

**Figure 1.** Researcher and student average rating of different sustainability reports (*n* = 49).

Results from Figure 1 indicate that students are not able to use the grid to differentiate between different levels of measurement maturity when researcher reporting is used as a reference. The student rating is similar over the entire range of performance. The overall correlation between researcher and student rating is poor, see also Figure 2.

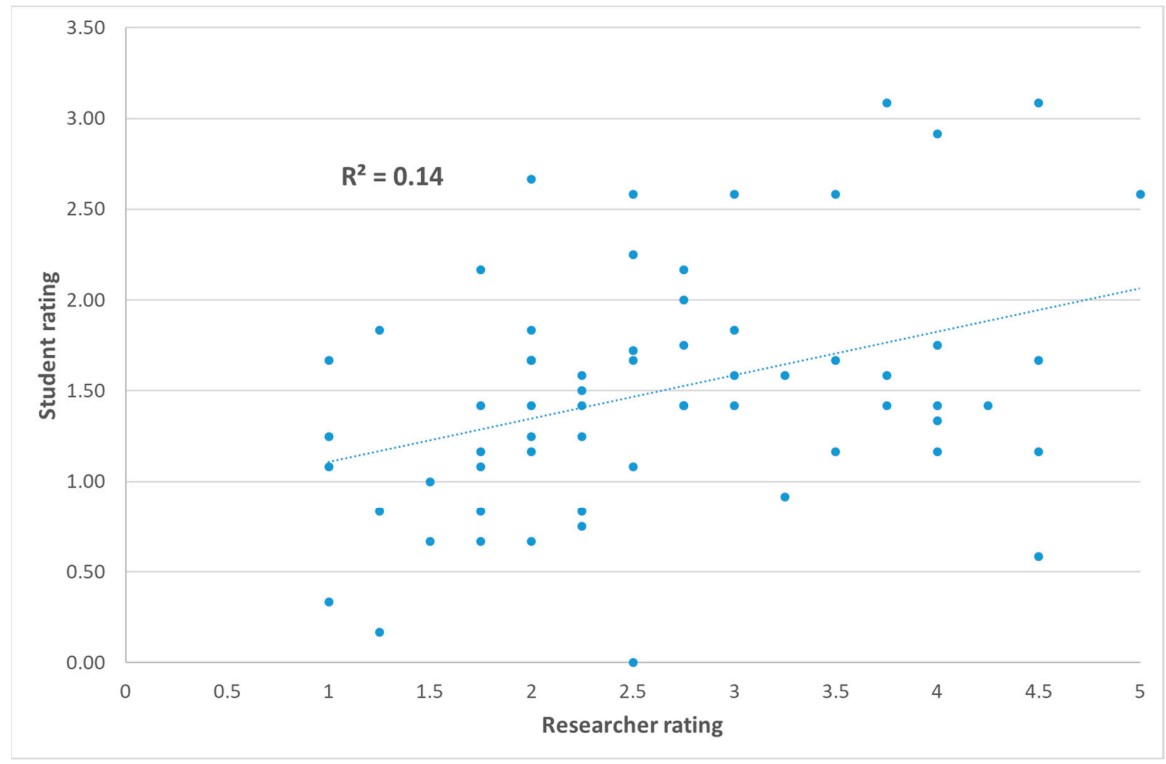

**Figure 2.** Correlation between researcher average and student rating of different sustainability reports (*n* = 64).

In Figure 3 researcher rating confidence intervals have been calculated. The average standard deviation between the researcher rating has been calculated to 0.466. Two standard deviations (2 s) have been used to calculate 95% confidence intervals. The 2 s have been assessed as 0.93. The current variation between researcher rating is still too high to enable an accurate rating of maturity reports. Based on Figure 3 the linear average confidence interval still indicates that even distinguishing between the lowest and highest performers could be put into question.

The rating of individual companies performed should be taken as indicative and relates only to the measurement maturity as defined per the grid.

Figures 1 and 2 indicate some outliers which could be students that have not taken the task seriously and that have responded randomly. In order to test this hypothesis, the difference of the average researcher and student rating was calculated and correlated with the student score in the examination. The hypothesis was that the higher the student score the smaller the difference in rating.

Results in Figure 4 are non-conclusive. There is an indication that a higher student score reduces the variability of the difference and that it could possibly reduce it. These results are similar to those found by Isaksson and Cöster [6].

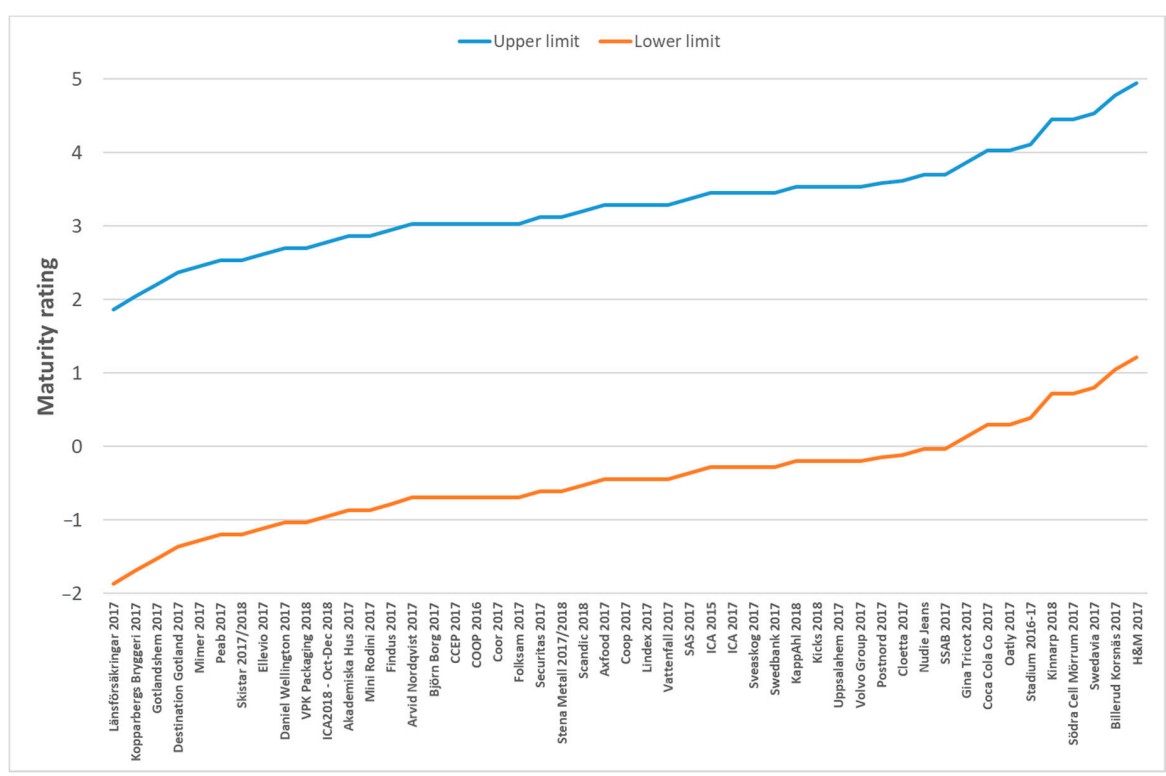

**Figure 3.** Researcher rating and average of reports.

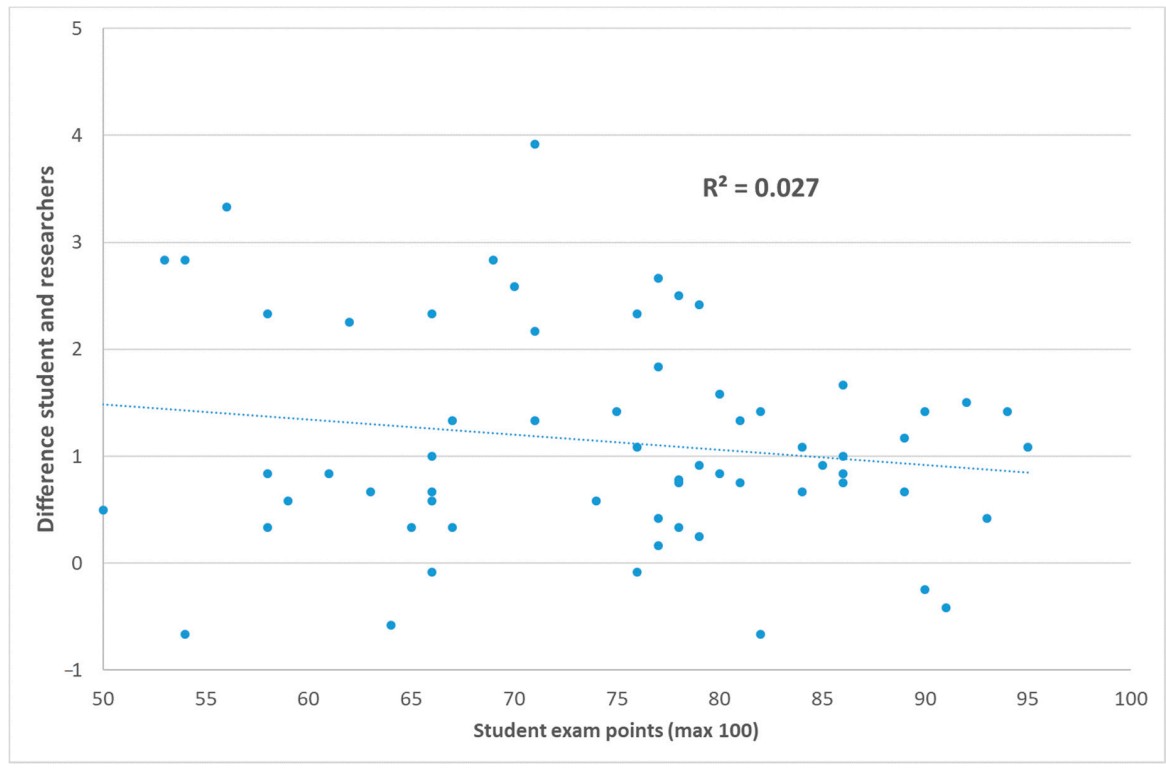

**Figure 4.** Difference between student and researcher rating correlated with exam points (*n* = 64).

### 5.4. SRMG Modification

During the work of analyzing sustainability reports using the maturity grid, a number of problems and areas of improvement surfaced.

With the purpose of seeing if the assessment could be improved, five assessments with large variations were studied by the researchers in a consensus discussion with the purpose of finding out causes for differences in the assessments. Different views could quickly be sorted out and input was received for clarifications in the maturity grid. Generally, it seems that an assessment done using a consensus discussion should improve the accuracy of the rating.

The logic of the maturity grid is based on doing the right thing right. The proposed modifications in Table 4 compared to Table 1 are with focus on increased clarity. The two main criteria of doing the right thing and doing it right in Table 1 have been divided into three parts in Table 4 with the purpose of clarification. For doing the right thing stakeholder identification and stakeholder needs identification have been separated. This is due to observations where it has been noted that relevant stakeholders have been identified but not necessarily the relevant needs. For doing the right thing right, indicators and targets have been separated for greater clarity. For better readability the grid axes have been shifted showings levels vertically. For readability the requirement for data presented has been increased to seven years or seven recordings. In order to establish a trend, seven values as commonly required as defined in Quality Management and Statistical Process Control [38].

Table 4. Proposed revised maturity grid.

| Level | Doing the Right Thing | | | Doing the Right Thing the Right Way | | |
|---|---|---|---|---|---|---|
| | Value Chain | Stakeholder Identification in Value Chain (People and Planet) | Stakeholder Needs Identification and Understanding Sustainability | Sustainability Performance Indicators | Targets for Sustainability | Readability |
| 0 | Not identified | Not identified | Stakeholder needs not identified | No clear indicators | No clear targets | Results mainly in text with no focus on layout |
| 1 | Implied but not clearly defined | Identified but not clearly defined | Stakeholder needs identified but not clearly defined | Some indicators | Some targets | Focus on layout highlighting main performance. Easy to find and easy to read report content. |
| 2 | Defined from cradle to grave/cradle | Clearly defined | Clearly defined. Organization demonstrates understanding of sustainability | Main indicators for defined sustainability performance | Targets for identified indicators. | Easy to read sustainability report/section presenting main results in Tables or Graphs |
| 3 | Clearly visualized | The UN SDGs or/and the Planetary Boundaries have been used to identify stakeholders | There is a materiality analysis identifying main stakeholder needs | Main indicators for sustainability are based on a materiality analysis | Targets for main indicators identified in materiality analysis | Relevant performance indicators for at least three years including carbon emissions. |
| 4 | Clear interfaces for which parts reporting is done | Identified stakeholders include climate (atmosphere) and other relevant stakeholders in value chain | The main stakeholder needs are expressed and organizational sustainability criteria are defined including climate effects | There are absolute and relative performance indicators describing main company sustainability performance including carbon emissions | Targets for main sustainability performance are defined based on external requirements including targets for carbon emissions | Presentation highlighting relevant results for main indicators in easy to understand graphical form combining trend and target for at least five years. |
| 5 | Clear that main sustainability impacts are reported for entire value chain | Clear and well-motivated priorities for all relevant stakeholders in value chain | Clear and well-motivated priorities for all relevant stakeholders needs | Clear indicators for all relevant and prioritized stakeholder needs | Clear externally based targets for all relevant and prioritized stakeholder needs | Extensive report with executive summary presenting the situation and performance over time for all relevant KPIs and targets in an easy to read manner for at least seven years |
| Rating | | | | | | |
| Average | | | | | | |

## 6. Discussion

The purpose of this paper is to present and test a maturity grid for sustainability reports assessment that enables critical stakeholder needs analysis of sustainability reports.

Another purpose is to use the grid for a first assessment of reporting sustainability maturity. The proposed grid has been used on several university courses in Uppsala University, Sweden, in addition to the one reported above. A common comment from students has been that it simplifies a critical analysis. There have also been comments that the grid is ambiguous and that the levels are difficult to determine.

An important shortcoming of the SRMG is the lack of validation of the results in comparison with other sustainability ratings. The scale in the grid is based on reasoning that identifies People and Planet as the main stakeholders and then further identifies the main stakeholder needs based on the Planetary Boundaries Framework and the UN Sustainable Development Goals. The proposed maturity grid in Table 4 still needs to be compared with other sustainability measurement maturity frameworks.

Another challenge that still remain is the lack of accuracy in assessing the reporting maturity. The research indicates that lower variation of rating could be achieved by a work process that combines individual rating and consensus discussion in a rating group. Further, remaining ambiguities in interpreting the grid text should be reduced.

In spite of these remaining problems, there is a clear indication that average performance reported in Table 3 ranging from 1.2 for the lowest researcher to 2.5 for the student average on a scale going to 5 is low.

The SRMG in Table 1 has been applied on sustainability reports published by Swedish companies, many of them being large international company groups. Sweden is by RobecoSam (2019) assessed as the world's leading country for sustainability [39]. This implies that its companies also could be expected to excel in sustainability actions and sustainability reporting. With this as a background the results presented in this paper indicate that global sustainability reporting still has a long way to go. Reasons for this may be vague organizational definitions of the concept of sustainability and hereby also unclear perception if organizations are doing the right things right.

Today the triple bottom line view on sustainability, for example represented by the UN SDGs 2030, are by many mechanistically used as an answer to the question what sustainable development addresses. For an organization that may be a too wide frame of analysis. No organization today would probably say that they are not in approval of sustainable development, but that is not enough. An organization needs to go beyond the overarching TBL view and identify what sustainable development is to them and how they contribute to it. When doing that, there may be a risk that work of such analysis is dumped in the lap of a sustainability department and/or on a chief sustainability officer, when it rather should be a driver for strategy development of the organization.

Still, when trying to identify what sustainability is to an organization, it is proper to start asking if it is doing the right things? That question leads to a need for identifying the value chain of the organization. As the maturity grid in Tables 1 and 4 illustrates, it can go from not having identified the value chain at all to being clear about how main sustainability impacts are reported for the entire value chain. Such an approach may enable the organization to critically analyze, discuss, and decide what the borders of the value chain are and the interfaces it has with for example subcontractors and customers. A challenge here is maybe to be bold enough and identify the whole value chain, and not just the parts that the organization directly controls. For example, if you are a retailer of clothes, working conditions at factories where these are produced as well as how raw materials such as cotton is grown, harvested, and refined should be a part of your value chain analysis. Results also indicate that this is the case. In, e.g., the large clothes retailer H&M sustainability report, which is part of the study, they describe all its upstream activities from raw materials and clothes production even if the company itself does not own any production facilities. Not doing this would most likely have damaged the brand and reduced sales. Generally, it seems that the clothing industry has a better coverage of the value chain than the

average company. It could be that business to business companies with little public scrutiny of their value chain might have a lower sustainability reporting maturity.

As the maturity grid implies, the other part of doing the right thing is to identify the stakeholders to the organization and the needs that these may have. When doing that, there is a chance that the analysis ends up in identification of only human stakeholders. That is not enough though, as the analysis needs to include the organization's influence on the greater stakeholder—the planet that humans live and thrive upon. Environmental issues and the limits to human and business activities are well described in the Planetary Boundaries framework [21]. We could, e.g., view the atmosphere as a stakeholder. If we do not treat it well the services it provides humanity could change, or in worst cases cease to exist. As a stakeholder it has requirements such as not increasing the amount of greenhouse gases and not producing chemicals that destroy the ozone layer. In a similar way we can view the biosphere as a stakeholder [10]. We need a number of ecosystem services which could be severely impaired by loss of biodiversity on land and in seas. All companies should understand which the impacts of their value chain are on areas such as climate, loss of biodiversity, and poverty [10]. This provides challenges in understanding the effects in the value chain and in delimiting the extent of involvement in value chain activities upstream and downstream from the business. If the value chain analysis has been essential that may give guidance to the stakeholder analysis. It can help the organization to come up with answers to questions regarding how various stakeholders are influenced by value chain activities.

The other part of the maturity grid in Table 4 implies that doing the right thing is not enough, as long as the activities in a value chain are not properly assessed and controlled. In other words, we need to check if things are done right. Here it is crucial that relevant absolute performance measurements are combined with sustainability key ratios and that these are grounded in the value chain activities. Absolute values, such as the total of greenhouse gas emissions, present the severity of the problem. Ratios such as sales per carbon emissions provide organizational comparability. These key ratios also have to be matched to sustainability targets that enable the organization to identify and discuss if or if not, sustainability actions are meeting the organizational targets. If that is done properly and systematically, there is a chance that the sustainability measures and targets can become an integrated part of the organizational control system. Already in the early 1990s, researchers advocated a need for a more nuanced organizational control that highlights not only financial performance but also other more intangible values [40–43]. Hereby the control systems may enable descriptive analysis of organizational performance, as well becoming a driver for strategic renewal. From a sustainability perspective that kind of renewal may include re-design of existing value chain and value creation activities.

According to Figures 1–3 indications are that maturity ratings vary and are individually based. This introduces bias in the results, but is a phenome highlighted by various authors within the field of maturity measurements, like Willems et al. [44] state that individual background affects the rating of maturity. Crosby [27] deems that when using his maturity grid, it is important that the organization or part of it, which is going to be evaluated, is evaluated by more than one person in order to visualize different views. In order to overcoming the problem of different persons from different functions not knowing or understanding all relevant information, Berg et al. [45] deem that an approach involving work-shop format when assessing maturity in comparison to individual assessment has advantages. The problem of being precise when judging the current level of maturity is discussed by Hammer [46], since organizational behaviors can be a mix of for instance two levels of maturity. Here Hammer [46] advocates an approach of marking the levels of maturity in the maturity model with the colors green, yellow, or red. Green is equal to that the statements are largely true (at least 80% correct), yellow between 20% and 80% correct, and red largely untrue, i.e., less than 20% correct. The green cells will then indicate the enablers, i.e., the level of maturity which the organization has passed, the yellow cells indicate the cells where the organization has a lot of work to do, and the red ones represents obstacles for evolving maturity [46].

## 7. Conclusions

If organizations wish to be transparent with their sustainability work, the sustainability report is a critical channel for documentation and information. The SMRG (see Table 4) that is presented in this paper enables an organization to identify if they report the right things based on a value chain and stakeholder analysis. Furthermore, if they are doing it right through the use of well-grounded KPIs and sustainability targets. This will in turn also enable them to communicate their sustainability efforts in a more precise and consistent way.

From the stakeholder perspective sustainability reports based on the criteria's in the SMRG are probably easier to interpret and critically analyze, as is suggested by multiple student feedback. It also enables stakeholders to compare content of various reports, and hereby, e.g., benchmark these towards each other.

Finally, the SMRG may increase transparency and readability by influencing the sustainability report process in an organization. If the maturity of these reports hereby increases, they may also become an important part of the strategic dialogue in the organization regarding how and in which ways it contributes to sustainable development.

The originality of this paper is primarily in using the outside in approach on main stakeholder needs and using this to identify core issues for sustainability. This is in strong contrast to most sustainability measurement maturity assessments, which are based upon an inside out view adding different "good" things to form a sustainability rating.

**Author Contributions:** Conceptualization M.C. and R.I.; methodology, M.C., G.D., R.I.; software, R.I.; validation, M.C. and R.I.; formal analysis, M.C., G.D., R.I.; investigation, M.C., G.D., R.I.; resources, R.I.; data curation, G.D., R.I.; writing—original draft preparation, M.C., G.D., R.I.; writing—review and editing, M.C., G.D., R.I.; visualization, R.I.; supervision, M.C. and R.I.; project administration, R.I.; funding acquisition, R.I. All authors have read and agreed to the published version of the manuscript.

**Funding:** Partial funding was received from the Swedish Institute for Quality (SIQ).

**Conflicts of Interest:** The authors declare no conflict of interest.

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
