# Peer review of "Are They Reporting the Right Thing and Are They Doing It Right?—A Measurement Maturity Grid for Evaluation of Sustainability Reports"

_sustainability, doi:10.3390/su122410393_

Round 1
Reviewer 1 Report
The paper discusses an interesting methodological proposal for analysing the quality of sustainability reporting. Despite the relevance of the topic, several critical points could be highlighted. The reviewer summarizes the aspects that deserve to be explored.
- The Authors specify that: “The grid does not intend to measure the level of company sustainability but only the sustainability reporting quality”. The sustainability reporting quality is the key point of the methodology (i.e the maturity grid) proposed. However, in the paper there is no precise definition of sustainability reporting quality. What exactly does the maturity grid measure? In the reviewer opinion, this point is unclear or unsatisfactory. The authors should clearly define, also through references to literature, what they mean by sustainability reporting quality. It’s the compliance at the GRI? It’s the process? Or it’s something else? If it’s the compliance, what are the variables and sub variables measured?
- Closely linked to this point there is also a further criticality related to the measurement of the four “statements” (Table 1). How was the score 0-5 assigned? Are there variables or sub-variables related to each of the four “statements” or is the choice of score totally subjective?
- It is not completely clear whether the final objective is to enhance the effectiveness of the grid, as a valid tool for evaluating the measure of sustainability reporting quality, or whether it is to test the alignment of scores between the two groups (researchers and students). In addition, If so many human resources are needed to apply the grid, is the tool efficient and can be replicated?
- Finally, can the results be influenced by the Swedish institutional context? Should a paragraph be added explaining the national context? Are there differences in sustainability reporting quality results by sector or by size?
Best regards and have a good work!
Author Response
We thank the reviewer for these constructive comments as a consequence we have met each of the comments as follows:
- The Authors specify that: “The grid does not intend to measure the level of company sustainability but only the sustainability reporting quality”. The sustainability reporting quality is the key point of the methodology (i.e the maturity grid) proposed. However, in the paper there is no precise definition of sustainability reporting quality. What exactly does the maturity grid measure? In the reviewer opinion, this point is unclear or unsatisfactory. The authors should clearly define, also through references to literature, what they mean by sustainability reporting quality. It’s the compliance at the GRI? It’s the process? Or it’s something else? If it’s the compliance, what are the variables and sub variables measured?
The statement has been clarified in the text. Sustainability reporting quality has been defined. See line 153 and forward.
- Closely linked to this point there is also a further criticality related to the measurement of the four “statements” (Table 1). How was the score 0-5 assigned? Are there variables or sub-variables related to each of the four “statements” or is the choice of score totally subjective?
The choice of the scale, which is based on previous work, has been explained and supported by arguments. See line 166 and forward
- It is not completely clear whether the final objective is to enhance the effectiveness of the grid, as a valid tool for evaluating the measure of sustainability reporting quality, or whether it is to test the alignment of scores between the two groups (researchers and students). In addition, If so many human resources are needed to apply the grid, is the tool efficient and can be replicated?
We write that: ”The purpose of this paper is therefore to present and test a maturity grid for sustainability reports assessment that enables critical stakeholder needs analysis of sustainability reports”. One way of testing is to see if students and to some extent more experts on the matter get the same results. The final objective is to enhance the effectiveness of the grid. We have justified the human effort needed in the discussion. With improvements our expectation is that the work done is justified by learning and improvements.
- Finally, can the results be influenced by the Swedish institutional context? Should a paragraph be added explaining the national context? Are there differences in sustainability reporting quality results by sector or by size?
Interesting question. We have not looked at differences in sectors. It can be assumed that those with Business to Customer have felt stronger requirements and therefore have a higher maturity. But, this is work for later research. We have argued that Sweden as a sustainability leader should be an expected benchmark. This is only a hypothesis and could be studied later.
Additionally we have also conducted some clarifications in the text and made some minor improvements regarding the language.
Best regards,
The Authors
Reviewer 2 Report
The article is an interesting one as far as the concept of Crosby's maturity grid is concerned. The analysis performed by the authors is an interesting and topical one, as much as each organization or company is fighting fiercely in order to elaborate and present the sustainability reports. Whether they choose the right sustainability indicators or a clearer specification is needed to highlight certain issues of major importance to stakeholders.
The literature clearly describes the evolution of the concept of the maturity grid for evaluating sustainability reports but also the SRMG proposal in Table 1. Here I would suggest the authors to use the same font as in the rest of the article text to keep its format.
Line 187 specifies figure 1, but figure 1 only appears on line 263. I don't think it refers to the figure on line 263! I suggest the authors to clarify this mentioned issue. The research methodology specifies the method inspired by IAR but the authors do not say what it consists of. The reader cannot clearly identify the model compared to the one not proposed by the authors. I suggest the authors clarify this issue.
In the results, I suggest to the authors that table 2 be adjusted so that it can fit on a single page, as in the case of tables 3 and 4. The conclusions are clear but the authors do not specify the possible limits of their research.
Author Response
We thank the reviewer for these constructive comments as a consequence we have met each of the comments as follows:
- The article is an interesting one as far as the concept of Crosby's maturity grid is concerned. The analysis performed by the authors is an interesting and topical one, as much as each organization or company is fighting fiercely in order to elaborate and present the sustainability reports. Whether they choose the right sustainability indicators or a clearer specification is needed to highlight certain issues of major importance to stakeholders.
We have rewritten the text justifying the chosen maturity criteria. See line 157 and forward
- The literature clearly describes the evolution of the concept of the maturity grid for evaluating sustainability reports but also the SRMG proposal in Table 1. Here I would suggest the authors to use the same font as in the rest of the article text to keep its format.
Font format in Table 1 has been changed
- Line 187 specifies figure 1, but figure 1 only appears on line 263. I don't think it refers to the figure on line 263! I suggest the authors to clarify this mentioned issue.
On line 187 figure 1 has been changed to table 1. The same change has been conducted on line 208. (After improvements of the document, line 187 and line 208 are now line 230 and line 250 in the revised document)
- The research methodology specifies the method inspired by IAR but the authors do not say what it consists of. The reader cannot clearly identify the model compared to the one not proposed by the authors. I suggest the authors clarify this issue.
The use of IAR has been elaborated, see line 225 and forward
- In the results, I suggest to the authors that table 2 be adjusted so that it can fit on a single page, as in the case of tables 3 and 4.
The column width of table 2 has been adjusted so that the table becomes shorter. But the number of items in the table are greater than the number of rows on a journal side. Here we also hope the journal support for forwards
- The conclusions are clear but the authors do not specify the possible limits of their research.
Limits of research have been added, see line 440 and forward
Additionally we have also conducted some clarifications in the text and made some minor improvements regarding the language.
Best regards,
The Authors
Reviewer 3 Report
This article presents an element of originality regarding the treatment of the maturity grid of the measurement of sustainability ratios.
I noticed that from the point of view of the research methodology, the method invoked as a starting point in the development of the authors' model is not presented but only specified. I suggest the authors present this method based on Crosby's evaluation grid.
From my point of view tables 2, 3 and 4 should be slightly arranged to fit the page (on a single page). I suggest the authors do this. Also figure 1 is mentioned in the text but spaced as a presentation. I suggest the authors synchronize this so that the reader has a better image of the presentation.
In the conclusions, the authors do not mention the limits of their research. I suggest the authors explain this and insist on the originality of their article.
Author Response
We thank the reviewer for these constructive comments as a consequence we have met each of the comments as follows:
- I noticed that from the point of view of the research methodology, the method invoked as a starting point in the development of the authors' model is not presented but only specified. I suggest the authors present this method based on Crosby's evaluation grid.
We have elaborated the research methodology, see line 143 and forward
- From my point of view tables 2, 3 and 4 should be slightly arranged to fit the page (on a single page). I suggest the authors do this.
We have tried with some improvements, but there are still things remaining, which we hope the journal support for.
- Also figure 1 is mentioned in the text but spaced as a presentation. I suggest the authors synchronize this so that the reader has a better image of the presentation.
In two places in the text, Table 1 has wrongly been referred to as Figure 1, which has been changed. After improvements of the document, the changes are now present on line 230 and line 250 in the revised document
- In the conclusions, the authors do not mention the limits of their research.
We have added some text on limits to research, see line 440 and forward
- I suggest the authors explain this and insist on the originality of their article.
We have added comments on the originality of the paper, see row 472 and forward
Additionally we have also conducted some clarifications in the text and made some minor improvements regarding the language.
Best regards,
The Authors
Round 2
Reviewer 1 Report
I thank the authors for their revision work. The interventions are in line with the requests and are satisfactory.
I wish the authors a good continuation in the research work.
Best regards